# Fermented Foods, Health and the Gut Microbiome

**DOI:** 10.3390/nu14071527

**Published:** 2022-04-06

**Authors:** Natasha K. Leeuwendaal, Catherine Stanton, Paul W. O’Toole, Tom P. Beresford

**Affiliations:** 1Teagasc Food Research Centre, P61 C996 Cork, Ireland; 109311718@umail.ucc.ie (N.K.L.); catherine.stanton@teagasc.ie (C.S.); 2APC Microbiome Ireland, University College Cork, T12 K8AF Cork, Ireland; pwotoole@ucc.ie; 3School of Microbiology, University College Cork, T12 K8AF Cork, Ireland

**Keywords:** fermented foods, food microbiota, diet, gut microbiome

## Abstract

Fermented foods have been a part of human diet for almost 10,000 years, and their level of diversity in the 21st century is substantial. The health benefits of fermented foods have been intensively investigated; identification of bioactive peptides and microbial metabolites in fermented foods that can positively affect human health has consolidated this interest. Each fermented food typically hosts a distinct population of microorganisms. Once ingested, nutrients and microorganisms from fermented foods may survive to interact with the gut microbiome, which can now be resolved at the species and strain level by metagenomics. Transient or long-term colonization of the gut by fermented food strains or impacts of fermented foods on indigenous gut microbes can therefore be determined. This review considers the primary food fermentation pathways and microorganisms involved, the potential health benefits, and the ability of these foodstuffs to impact the gut microbiome once ingested either through compounds produced during the fermentation process or through interactions with microorganisms from the fermented food that are capable of surviving in the gastro-intestinal transit. This review clearly shows that fermented foods can affect the gut microbiome in both the short and long term, and should be considered an important element of the human diet.

## 1. Introduction

Fermented foods have been a component of the human diet from ancient times. Among the earliest evidence for the deliberate application of fermentation has been found in pottery vessels discovered in China dating from 7000 BC that were used to ferment rice, honey, and fruit [1]. However, it is likely that inadvertent production and consumption of fermented foods significantly predate this, as foods must have undergone spontaneous fermentation during storage [2]. While microorganisms, first discovered in the 1670s by Antoni van Leeuwenhoek, originally received much attention as agents of food spoilage and disease, useful applications, including their ability to produce antibiotics against pathogenic bacteria and to positively impact human health, were soon discovered [3]. However, probably the most important function that microbes have played throughout human history is their involvement in food preservation via fermentation [3,4]. Fermentation is the process whereby alcohols, carbon dioxide, and/or organic acids are produced by microorganisms, primarily from sugars and under mostly anaerobic conditions, for production of energy [5]. The accumulation of alcohol and organic acids and the associated increase in acidity of the food substrates inhibits the growth of other microorganisms and the activity of enzymes in the food system, thus reducing the rate of spoilage and resulting in foods with extended shelf-life.

However, while the primary function of food fermentation revolves around enhanced food safety and extended shelf-life, fermented foods have become associated with health benefits as well. One of the earliest proponents of this hypothesis was Élie Metchnikoff, who was interested in the potential of food in promoting the elongation of life. In Metchnikoff’s essay entitled ‘Lactic Acid and Putrefaction’, he attributes the long lives of Bulgarian peasants to the staple foods of the country at the time, in particular soured milk. In his experiments, he found that lactic acid bacteria (LAB) cultures in the fermented food produced ‘disinfecting bodies’ beneficial to their human host [6]. Thus, fermented foods have been linked positively with human health beginning in the early 1900s; the mechanisms subsequently postulated by which these foods can benefit health include one or a combination of the following: (i) the direct nutritional value of fermented foods, including bioactive compounds, produced as a consequence of the fermentation process; (ii) provision of nutrients to promote growth of indigenous gut microbes; and (iii) the capacity of the microbes in fermented foods to survive gastric transit and to either become a component of the gut microbiome or to inhibit/compete with existing members of the gut microbiome.

Consequently, researchers have proposed that fermented foods be included as part of dietary guidelines. Chilton and colleagues argue that due to their long role as part of the human diet and their established and emerging nutritional and therapeutic benefits (which will be discussed in detail below), brought about in part by their own microbiome, they deserve to be considered for regular consumption and to be included in food consumption guidelines [5]. In addition, probiotic bacteria, defined as ‘live microorganisms, which when consumed in adequate amounts confer a health benefit on the host’, can be a part of the fermented food microbiome [7,8]. Many fermented foods contain bacteria with probiotic potential, either added during the manufacturing process or adventitious bacteria, such as non-starter lactic acid bacteria (NSLAB) found in cheese, that are able to grow and thrive in fermented foods [9]. ‘Functional Foods’ is a legal term coined in Japan that was defined in Europe in 1999 as a food that has “satisfactorily demonstrated to affect beneficially one or more target functions in the body, beyond adequate nutritional effects, in a way that is relevant to either an improved state of health and well-being and/or reduction of risk of disease” [10,11]. The potential for fermented foods to positively impact the gut microbiome suggests that further research is required on this topic in order to establish whether fermented foods can in fact be defined as functional foods.

In this review, a brief introduction to fermented foods and their potential health benefits is followed by a discussion of the ability of fermented foods to impact the gut microbiome, either through fermented food communities surviving gastric transit and becoming established transiently or permanently as part of the gut microbiome, or through provision of nutrients that support the growth of indigenous gut microbiota; in addition, we explore the potential nutritional and health benefits associated with such interactions.

## 2. Fermented Food

The earliest evidence of food fermentation dates from the Neolithic period, ~7000 BC, in China [1], a period that coincides with the development of agriculture and thus of seasons in which an abundance of food was available followed by periods of relative scarcity. Cultivation of crops started in the Fertile Crescent of Southwest Asia ~9000 BC, and there is evidence that goat, sheep, and cattle were domesticated by 7000 BC [12]. Domestication of animals provided a source of meat and milk, which in its raw form has a very short storage time before spoiling. However, rudimentary processes to convert milk to cheese by fermentation had emerged by 6500 BC, the key benefit of which was an extended shelf life; a mechanism was thereby found to save food available in a time of plenty for one of relative scarcity. The process of making cheese had the additional advantages of (i) concentrating the fat, protein, and mineral content of the milk thus producing an energy-dense and nutritious food, and (ii) removing lactose, making the cheese accessible to adults, who at that time were mostly lactose intolerant and could not consume milk post-infancy [12].

The process of fermentation has been exploited by almost all peoples and has been applied to plant materials (including fruits, seeds, tubers and other vegetative and non-vegetative materials) and animal materials (including meat, milk, fish, and eggs), reflecting the base foods available in different regions [2,13].

### 2.1. Diversity of Fermented Food Types

It is difficult to definitively establish the number of fermented foods produced globally; most estimates suggest that there are in excess of 5000 different kinds [14]. However, when local and regional variations are considered, this figure is likely to expand significantly. For example, different classification schemes can be used to classify cheese, and while certain approaches suggest as few as eighteen primary cheese types, when local modifications based on variations in procedures, microorganisms, and milk types are taken into account it is generally accepted that in excess of 1000 cheese varieties are available globally [15]. Different approaches have been used to characterize or group fermented foods, the most common of which are based on the raw/non-fermented food substrate, resulting in groupings of fermented foods made from (1) cereals, (2) vegetables, (3) legumes, (4) roots/tubers, (5) milk, (6) meat, (7) fish, (8) alcoholic beverages, and (9) miscellaneous [16].

As a consequence of their ancient origins, there is a strong regional bias in the distribution and popularity of different fermented foods. For example, throughout much of East and South Asia and the southern regions of India fermented legumes, in particular soyabeans, as well as vegetables, fish, and meat are common components of the diet. In West Asia, Northern India, Europe, and North America, fermented cereals, including wheat, barley, and oats, are used to make bread, and fermented dairy and meat products are more common. In Africa and South America fermented seeds, including sorghum, maize, millet, cassava, and legumes, are an important component of the diet, as are fermented milk and meat products [16]. A recent review by Tamang and colleagues (2020) expertly demonstrates these differences in the origins of fermented foods around the world [2].

### 2.2. Primary Food Fermentation Pathways

Fermentation is a process whereby substrates are converted to alcohols, carbon dioxide, and/or organic acids, predominantly anaerobically [5]. From the microorganisms’ perspective, the purpose of fermentation is energy production [17]. Microbes adapted to diverse environments exhibit a range of different fermentation pathways; the pathways most relevant to fermented foods are listed in Table 1.

The substrate for both lactic acid and ethanol fermentation consists of the sugars available in the food material, for example (though not limited to) lactose in milk and glucose from the breakdown of starch in plant products; the products of both pathways, in addition to influencing the sensory characteristics of the fermented food, play a critical role in extending its shelf life, as the accumulated lactic acid or ethanol inhibits the growth of undesirable spoilage and/or pathogenic microorganisms. Propionic or acetic acid fermentation is effectively secondary fermentation, the substrate being the product of lactic or ethanol fermentation, respectively. This contributes to extended shelf life and is critically important in impacting the sensory characteristics of the final product. Citric and malolactic fermentations use substrates available in the raw food material, which are converted into products that impact the sensory characteristics of the fermented food. It is important to note that during the food fermentation process, the microorganisms present can release amino acids and encrypted bioactive peptides from proteins, convert fats to more healthy formats such as conjugated linoleic acid, and produce a wide diversity of metabolites, including short chain fatty acids (SCFA), vitamins, exopolysaccharides, and gamma-aminobutyric acid; these impact the sensory characteristics and health-promoting potential of the final fermented food product [24].

The lactic acid fermentation pathway is arguably the most important fermentation pathway in terms of fermented foods. It is the basis of all dairy fermentation as well as the majority of fermentation processes using plant or animal source materials. The bacteria responsible are LAB, which are grouped depending on whether they are homo-, hetero-, or facultative fermenters [17]. While homofermentation solely produces two moles of lactate per one mole of glucose, heterofermentation produces ethanol and carbon dioxide in conjunction with lactate [17]. This is due to homofermentative LAB harbouring the aldolase enzyme and heterofermentative LAB using phosphoketolase, resulting in divergent catabolic pathways [25]. Facultative LAB can ferment by either the homo- or the heterofermentation pathway depending on environmental conditions and substrate availability. The homofermentative LAB most commonly associated with fermented foods include *Lactococcus lactis*, *Streptococcus thermophilus*, homofermentative Lactobacillaceae including *Lactobacillus delbrueckii* subspecies *bulgaricus*, *Lactobacillus acidophilus*, and *Lactobacillus helveticus*, and members of the *Pediococcus* and *Enterococcus* genera [26]. Heterofermentative LAB includes *Leuconostoc* spp. and several lactobacilli, including *Fructilactobacillus sanfranciscensis*, *Levilactobacillus brevis*, *Limosilactobacillus fermentum*, and *Limosilactobacillus reuteri*. The most commonly encountered facultative LAB include *Lactiplantibacillus plantarum*, *Lacticaseibacillus casei*, and *Latilactobacillus curvatus* [26].

Ethanol fermentation is responsible for the production of wines from fermented fruits, beers from cereals such as wheat, barley and rye, and other stronger liquors involving distillation following fermentation such as sake from rice, vodka from potatoes, and rum from sugar cane [27]. The organism most commonly responsible for ethanol fermentation is the yeast *Saccharomyces cerevisiae*, although *Zymomonas mobilis* (a gram-negative bacterium) is used for, among other products, the production of the fermented Mexican beverage Pulque [17,28]. While a mole of glucose is converted to two moles of both ethanol and carbon dioxide regardless of whether yeast or *Zymomonas mobilis* is used, the pathways differ [25]. While both involve the enzyme pyruvate decarboxylase and the intermediates pyruvate and acetaldehyde and both lead to the same metabolites, ethanol fermentation by yeasts yields two moles of ATP per mole of glucose while *Zymomonas* generates only one mole of ATP [25]. This difference is because yeasts utilize the Glycolytic pathway and *Zymomonas* uses a variation of this, the Entner–Doudoroff pathway [17]. It is worth noting that in fermented foods ethanol can be produced, albeit at low concentrations, by heterolactic fermentation [25]. Kefir, which can contain up to 2% ethanol, is a good example of this; the ethanol is produced by both the yeast and heterofermentative LAB present during the fermentative process [29].

Acetic acid is an important component in vinegar and is derived from fermenting alcoholic beverages such as wine, cider, and beer; the end product differs depending on the region in which it is produced and the alcoholic beverage used [30]. As discussed above, in alcoholic fermentations glucose is converted to ethanol, which is followed by oxidative fermentation to produce acetic acid [31]. Where homoacetic fermentation occurs, one mole of acetate is produced from each mole of ethanol consumed. Acetic Acid Bacteria (AAB) are responsible for this process [25]. In terms of commercial vinegar production, aerobic species of *Acetobacter* and *Komagataeibacter* are of particular interest due to their high tolerance of both the substrate (ethanol) and product (acetate) present in their growth media [31].

Emmental and other Swiss-type cheeses rely on the ability of Propionic Acid Bacteria (PAB) to ferment lactate to propionic acid, with the coproduction of acetate and CO_2_ giving rise to the characteristic flavour and holes or “eyes” associated with these cheese types [15]. Propionate can be produced through either the acrylate or methylmalonyl–CoA pathway, although both produce two moles of propionate, one mole of acetate, and one mole of CO_2_ for every three moles of lactate catabolized [25]. PAB used in the production of Swiss-type cheeses utilize the latter methylmalonyl–CoA pathway [15].

Citrate fermentation is an important metabolic pathway for certain fermented foods, in particular in the dairy sector [32]. This fermentation can be undertaken by a limited range of homo- and heterolactic LAB, the most important being *L. lactis* subsp. *lactis* biovar *diacetylactis* and certain *Leuconostoc* species, respectively. The fermentation pathway and the end products produced vary depending on the LAB involved as well as environmental factors, including the availability of fermentable carbohydrate and the pH. Potential end products include acetate, formate, ethanol, 2,3-butanediol, diacetyl, acetoin, carbon dioxide, and lactate [33], of which diacetyl and acetoin are known to exhibit nutty and buttery aromatic notes. Citric acid fermentation is thus important in the production of a number of dairy products, such as cottage cheese, where diacetyl confers a buttery aroma, as well as in Gouda and other cheese varieties where in addition to impacting on aroma notes, the production of carbon dioxide is responsible for eye formation [33].

Lactic acid bacteria are responsible for the malolactic fermentation process that is involved in the production of wine as well; this predominantly takes place after alcohol fermentation by yeast [34]. This reaction converts L-malic acid to L-lactic acid and carbon dioxide and results in the production of other flavour compounds of interest such as diacetyl [35], leads to deacidification of the wine, and promotes microbial stability [36]. The predominant LAB associated with this process is *Oenococcus oeni*, although other Lactobacillaceae and *Pediococcus* species can be used [37].

### 2.3. Microbiome of Fermented Foods

The primary microorganisms responsible for the main fermentation pathways that occur in food are discussed above and are listed in Table 1. Traditionally, food fermentation has relied on the microbiota naturally occurring in the raw food material or obtained by transferring the microbiome from previously-fermented products. However, under modern large-scale commercial food production systems well-characterized or defined starter cultures are now widely used to ensure reproducible products capable of consistently meeting high consumer standards, and the safety status of these microbes are established to ensure the safety of consumers [38].

However, even under modern food processing systems most of the ingredients used in the preparation of fermented foods and the processing equipment used in their manufacture are not sterilized; thus, in addition to known starter microbiota most fermented foods and beverages contain an indigenous microbiome, potentially consisting of a wide variety of microorganisms. Traditional microbiological methods demonstrate the existence of these indigenous microorganisms, often referred to as Non-Starter organisms; however, with the development and widespread application of affordable and reliable high-throughput DNA sequencing technologies numerous fermented foods have undergone metagenomic screening, revealing highly diverse and previously unrecognized populations of adventitious Non-Starter microorganisms [4,16,39,40]. The diversity of microorganisms includes bacteria, yeast, and filamentous moulds. In addition to the bacteria listed in Table 1, species of *Arthrobacter*, *Bacillus*, *Bifidobacterium*, *Brachybacterium*, *Brevibacterium*, *Enterobacter*, *Hafnia*, *Haloanaerobium*, *Halobacterium*, *Halococcus*, *Klebsiella*, *Kocuria*, *Micrococcus*, *Pseudomonas*, and *Staphylococcus* have been recorded in different fermented foods [41]. The genera of yeast identified include *Brettanomyces*, *Candida*, *Cryptococcus*, *Debaryomyces*, *Dekkera*, *Galactomyces*, *Geotrichum*, *Hansenula*, *Hanseniaspora*, *Hyphopichia*, *Issatchenkia*, *Kazachstania*, *Kluyveromyces*, *Metschnikowia*, *Pichia*, *Rhodotorula*, *Rhodosporidium*, *Saccharomyces*, *Saccharomycodes*, *Saccharomycopsis*, *Schizosaccharomyces*, *Sporobolomyces*, *Torulaspora*, *Torulopsis*, *Trichosporon*, *Yarrowia*, and *Zygosaccharomyces*, while the filamentous moulds include *Actinomucor*, *Amylomyces*, *Aspergillus*, *Monascus*, *Mucor*, *Neurospora*, *Parcilomyces*, *Penicillium*, *Rhizopus*, and *Ustilago* [16]. By assessing the microbial diversity of fermented foods, a more reliable understanding of their fermentation potential can be determined, thus allowing for a more complete understanding of each fermented food and its potential to impact human health.

The diversity that exists among fermented food microbiomes is exemplified by cheese. The microbiome that develops in cheese during ripening arises from the milk and other ingredients used for cheese manufacture, the starter cultures added at the beginning of manufacture, and the cheesemaking and ripening room environments [42]. While the majority of cheeses are produced from cow’s milk, cheese from sheep, goats, buffalo, camels, and even donkeys is manufactured [43]. Genera common to all raw milk include *Lactococcus*, *Lactobacillus*, *Leuconostoc*, *Streptococcus*, and *Enterococcus*, although certain varieties are more strongly associated with one type or another, which impacts the microbiome of a final product [44,45,46]. For example, while the predominant genera in cow’s milk include *Pseudomonas*, *Bacillus*, *Lactococcus*, and *Acinetobacter*, the genera most common to raw camel’s milk include *Enterococcus*, *Lactococcus*, and *Pediococcus* [47,48]. The starters and adjunct cultures added to the milk at the beginning of the cheese-making process impact the food microbiome, with both bacterial and yeast/mold species sometimes being necessary for development of characteristic organoleptic properties [49]. For compilations of popular starter and adjunct cultures used in cheese preparation, González-González et al. (2022) and Bintsis (2021) have written comprehensive reviews on bacteria and yeast/mold cultures, respectively [50,51]. NSLAB are another important component of cheese; while they are present at low numbers immediately following cheese production (10^2^–10^3^ colony-forming units (CFU)/g), they proliferate and outcompete the starter cultures to become the dominant microbial populations during cheese ripening and are often responsible for the development of characteristic organoleptic properties [52]. Certain cheese microbiomes can have the added complexity of maintaining distinct microbial communities, such as smear cheese. Smear cheese is produced by rubbing or ‘smearing’ microorganisms onto the surface of the newly formed cheese, resulting in the development of a rind with a distinct flavour and a surface microbiome that differs greatly from the microbiome present in the cheese core [53,54]. Stilton, defined as a blue cheese due to veins of *Penicillium roqueforti* culture that emanate from the core outwards, has a very diverse microbiome that includes an outer rind that develops from different molds [55,56].

In addition, many fermented foods are eaten without further processing or preparation, and thus contain microbial populations of up to 10^8^ CFU/g; these can potentially gain entry to the human gastrointestinal tract, where they may interact with or become established as part of the gut microbiome [57,58].

## 3. Health Benefits of Fermented Foods

As discussed above, while fermented foods have a long history of safe use, there is a growing popular consensus that consumption of fermented foods results in positive health effects. Much of this is driven by popular observations that fermented foods in general use unprocessed raw ingredients, contain little or no added preservatives, colours or flavourings, and are made using long-established, sustainable, and in many cases traditional technologies. Consumers may be attracted to the concept that these are “live foods” containing natural and diverse microbiota.

### 3.1. Human Dietary Studies

It is incumbent upon the relevant research community to generate data to clarify and identify the benefits of such foods, if any exist. In support of this, a number of controlled human dietary studies have recently been undertaken which sustain these popular perceptions of health benefits [57]. These studies include investigations that revealed strong associations between weight management and consumption of fermented dairy products [59], reduced risk of cardiovascular disease, type 2 diabetes, and mortality associated with consumption of yoghurt [60,61,62,63], and enhanced glucose metabolism and reduced muscle soreness following acute resistance exercise as a consequence of consuming fermented milk [64]. Consumption of kimchi was linked to anti-diabetic and anti-obesity effects [65,66], while consumption of different fermented foods was associated with alterations in mood and brain activity [67,68,69] and in the gut microbiome [70]. However, reports have identified a lack of sufficient clinical trials, variation in the different fermented foods being investigated, and inconsistencies among ethnic groups in suggesting that more studies need to be undertaken in order to confirm the potential benefits of fermented food consumption [71].

### 3.2. Transformations in Food Arising from Fermentation

It well established that fermentation can enhance the digestibility of complex carbohydrates and proteins through the breakdown of starch to oligosaccharides and polypeptides to amino acids [72,73]. Fermentation allows for the destabilization of the casein micelle by bacteria present in milk, enhancing milk protein digestibility [24,74]. Fermentation, in particular with respect to cheese, facilitates the concentration of key nutrients through removal of water and enhances the bioavailability of calcium, which is important for skeletal health [75].

Additionally, fermentation can facilitate transformations in raw foods that allow these foods to be tolerated by consumers that are intolerant of the original raw product. A good example of this is the ability of lactose-intolerant individuals to consume fermented dairy products, in particular ripened cheeses such as Cheddar. The reason for this is that during fermentation and cheese ripening, the LAB metabolize the lactose, significantly reducing the level of lactose in the resulting fermented food product. In addition, the presence of the lactase enzyme produced by bacteria present in the fermented matrix can help to further remove any residual lactose during ingestion and digestion [76]. A similar example involves decrease in the concentration of anti-nutritional components in raw food, such as the partial eradication of harmful trypsin inhibitors during soy bean fermentation [77].

In addition, bioactive compounds can be produced through protein, lipid, and carbohydrate catabolism during the fermentation process, and can result from the range of microbial metabolites produced during the process [78]. Indeed, production of vitamins and antioxidants during food fermentation has been reported for many LAB species, in particular members of the newly-rebranded Lactobacillaceae family [79,80,81,82]. Bioactivities linked to lowering of blood pressure and cholesterol, improvement in metabolic syndromes, anti-cancer effects, and improvement in immune function have all been described [80]. Vitamins B7, B11, and B12 are produced in fermented dairy products by Lactobacillaceae (e.g., *Lactiplantibacillus plantarum*, *Lactobacillus delbrueckii*, *Limosilactobacillus reuteri*), *Propionibacterium*, *Bifidobacterium*, and several species of *Streptococcus* [83]. Folic Acid (B11) is necessary for development, and reproduction and prevents against certain disorders including some cancers and cardiovascular diseases, while many metabolic processes require B12 as a cofactor, including nucleic and amino acid metabolism [84]. Non-dairy fermented foods contain microbes which synthesize vitamins [29]. Shalgam is a Turkish fermented beverage consisting of black carrots and turnips; its resident microbiota consists predominantly of Lactobacillaceae along with *Leuconostoc* and *Pediococcus* [85]. This beverage is an abundant source of vitamins A, B, and C as well as other minerals and polyphenols [29,85]. Antioxidants exert beneficial functions in food and protect against the damaging effects of free radicals, and are produced in fermented foods by microbial esterases [79]. Kombucha, a fermented sweetened tea, contains many antioxidants with positive effects on health, including antagonistic effects towards progression of neurodegenerative diseases, diabetes, and certain cancers [29]. Many other fermented foods have been shown to positively contribute to different aspects of human health, such as kimchi, a fermented cabbage, which is capable of anti-atherogenic effects that are triggered by the active compound 3-(4′-Hydroxyl-3′,5′-dimethoxyphenyl) propionic acid (HDMPPA) [86].

### 3.3. Release of Bioactive Peptides

The release of bioactive peptides as a result of protein hydrolysis during fermentation has been investigated by multiple groups. A well-studied example of bioactive peptide release as a consequence of catabolism of protein involves angiotensin-1-converting enzyme (ACE)-inhibitory peptides, a group of peptides with hypertension-lowering capabilities [80]. LAB from both dairy and non-dairy sources can produce ACE-inhibiting peptides during fermentation of milk [87]. They are primarily produced in fermented dairy foods such as yogurt and cheese, although different casein molecules can be targeted depending on the starter LAB cultures used [88]. Two lactotripeptides, isoleucine–proline–proline (IPP) and valine–proline–proline (VPP), which are not digested and remain intact and able to cross the mucosal surface to confer their antihypertensive benefits, have received much attention [87]. For example, in one study, spontaneously hypertensive rats were fed a fermented milk beverage containing a *Lactobacillus helveticus* strain, a species known for its ability to release ACE-inhibitory peptides from milk protein, which significantly lowered blood pressure when compared with controls [89]. Human trials have yielded promising results as well. A meta-analysis of the published literature on human intervention trials confirmed the efficacy of IPP and VPP, including those in functional foods; Asian patients in particular reported a decrease in blood pressure which appeared to be independent of age, lactotripeptide dosage, length of treatment, or initial blood pressure readings at a level that was statistically significant [90]. ACE-inhibitory peptides have been identified in non-dairy products such as fermented meat sausages made using strains of *Latilactobacillus sakei* and *Latilactobacillus curvatus* [91].

### 3.4. Production of Exopolysaccharide

Many food fermentation microbes are capable of producing high molecular weight exopolysaccharides (EPS) from simple sugars present in the raw food product. EPS can be produced by *Zymomonas*, *Leuconostoc*, *Pediococcus* and *Streptococcus* species, and members of the Lactobacillaceae family [79]. Examples of EPS production include acetan, xanthan, and kefiran, several of which are important from a food manufacturing point of view. Xanthan is used for its desirable rheological characteristics when added to fermented milk products [92]. In terms of health, EPS-producing LAB have been found to have a role in immunomodulation, which can be either stimulatory or suppressive depending on various factors [93]. Dendritic cells bind to ingested microbial EPS via their pathogen recognition receptors through epithelial cell tight junctions, migrate with the EPS ‘antigen’ to other lymphoid tissues, and further communicate with other immunomodulatory cells such as Natural Killer cells, which can invoke inflammatory responses when appropriate [93]. EPS production in fermented foods has been examined in terms of cardiovascular disease (CVD) due to their ability to bind cholesterol, such as the ability of *Pediococcus*-produced β-glycans to lower serum cholesterol levels [93]. While prebiotics and probiotic bacteria are capable of lowering cholesterol through various mechanisms, EPS appears to lower cholesterol by binding bile (of which cholesterol is a constituent) from the intestines to the bacterial cell envelope, thus reducing bile reabsorption and recycling. Consequently, de novo synthesis of bile in the liver results in lowered serum cholesterol levels [94]. A paper in 2002 reported that EPS-producing starter cultures were able to bind the bile salt cholic acid, while another study in 2010 demonstrated the ability of EPS-producing bacteria to bind bile from liquid growth media and that the amount of EPS produced was correlated with a decrease in broth cholesterol levels [94,95]. An in vivo study investigating the ability of β-glucan-producing *Lacticaseibacillus paracasei* to lower serum cholesterol in mice found that a significant decrease was observed when compared to control mice that were not given the EPS-producing *Lacticaseibacillus* strain [96]. Additionally, this study showed that the gut microbiome was modified as a result of receiving the EPS-producing *Lacticaseibacillus paracasei* [96]. A similar study was performed on humans with elevated serum cholesterol levels who were fed fermented oat-based products containing an EPS-producing *Pediococcus damnosus* strain [97]. This study concurrently noted that a statistically significant reduction in total cholesterol was seen in the group who were given the fermented product, along with a significant increase in the relative abundance of faecal *Bifidobacterium* species (which are beneficial gut microbes) and total faecal bacterial counts [97]. Thus, fermented foods containing EPS can positively influence gut health.

## 4. Evidence of Fermented Foods That Modulate the Gut Microbiome

The ability of fermented foods to modulate the gut microbiome has been documented by several groups with varying degrees of success. Changes are generally recorded as overall shifts in microbial populations, and do not necessarily reflect the microbial content of the fermented foods in question. A recent study evaluated the effect of general fermented plant intake on microbial and metabolomic differences in consumers versus non-consumers (nearly 7000 participants total), and determined that beta diversity was significantly different between consumers versus non-consumers [70]. Microbiomes of consumers of such fermented products were associated with *Bacteroides* spp., *Pseudomonas* spp., *Dorea* spp., *Lachnospiraceae*, *Prevotella* spp., *Alistipes putredinis*, *Oscillospira* spp., *Enterobacteriaceae*, *Fusobacterium* spp., *Actinomyces* spp., *Achromobacter* spp., *Clostridium clostridioforme*, *Faecalibacterium prausnitzii*, *Bacteroides uniformis*, *Clostridiales*, and *Delftia* spp. [70]. In parallel, 115 separate participants who consumed fermented foods at various frequencies were investigated for a four-week duration, and it was found that microbes associated with the consumers of fermented foods consisted of both fermented food-associated microbes (e.g., *Lactobacillus acidophilus*, *Levilactobacillus brevis*, *Lactobacillus kefiranofaciens*, *Lentilactobacillus parabuchneri*, *Lactobacillus helveticus*, and *Latilactobacillus sakei*) and microbes not associated with fermented foods (including *Streptococcus dysgalactiae*, *Prevotella melaninogenica*, *Enorma massiliensis*, *Prevotella multiformis*, *Enterococcus cecorum*, and *Bacteroides paurosaccharolyticus*) [70]. In a similar fashion to the previous study, Wastyk and colleagues (2021) examined the effects of a diet rich in fermented foods (including fermented dairy products, vegetables and non-alcoholic beverages) on eighteen healthy adults over a period of seventeen weeks in parallel to a diet high in fibre [98]. The dietary intervention consisted of an initial four-week period whereby the quantity of fermented food in the diet was increased, followed by a six-week ‘maintenance’ period of very high fermented food intake, and closed with a ‘choice’ period of four weeks in which participants could maintain whatever level of fermented food intake they desired. The fermented food-rich diet resulted in an increase in alpha diversity of the gut microbiome that was not observed with the fibre diet. Interestingly, the increase in microbiome diversity remained during the designated ‘choice’ period despite intake being higher during the ‘maintenance’ period, with a strong relationship between time and diversity [98].

Different groups have reported changes in gut microbe populations following ingestion of fermented milk, although parameters vary between studies [99,100,101]. A fermented milk product with defined starters and adjuncts was able to significantly increase SCFA levels in vitro, especially butyrate; when administered to irritable bowel syndrome (IBS) sufferers, it led to a decrease in *Bilophila wadsworthia* (a so-called pathobiont, that is, a bacterium negatively associated with health) and an increase in two *Clostridiales* isolates (uncharacterized genera MGS203 and MGS126) known for butyrate production [99]. A separate study examined the effects of kefir on inflammatory bowel disease (IBD) patients; a significant increase in faecal Lactobacillaceae abundance was observed following consumption of the fermented milk for one month [101]. Zeng et al. (2021) administered either milk or kefir to a colorectal cancer mouse model and observed that the kefir group experienced gut microbiome perturbations that were absent in the milk counterparts, including a relative increase and decrease in probiotic-associated and pathogenic bacteria, respectively, as well as decreased *Firmicutes*/*Bacteroidetes* rand *Ascomycota*/*Basidiomycota* ratios at the phylum level [102]. Healthy females receiving symbiotic fermented milk experienced an increase in the relative abundance of faecal *Bacteroidetes* (in particular, the *Bacteroidaceae* and *Prevotellaceae* families) species and a decrease in *Firmicutes* abundance (in particular, the *Ruminococcaceae* and *Lachnospiraceae*) that reversed when the product was no longer being ingested [100]. This result is interesting, as the fermented product contained high levels of *Firmicutes* and yet a decrease in this phylum was observed, highlighting the varying and complex relationship between fermented foods and the gut microbiome. In contrast, Tillisch and colleagues (2013) reported no significant changes to the microbial composition of stool samples following ingestion of a probiotic-containing fermented milk [67]. Children infected with *Helicobacter pylori* exhibited lower stool *Bifidobacterium* than their healthy counterparts, which could be restored partially through ingestion of a probiotic yogurt that led to a significant increase in their stool *Bifidobacterium* spp./*Escherichia coli* ratio [103]. The effects of yogurt consumption on the gut microbiome of healthy subjects after 42 days was found to induce changes in overall composition and diversity, although this varied between individuals and no significant fluctuations were observed [104]. Firmesse and colleagues examined the effects of Camembert consumption in healthy subjects over a four-week period, focusing on faecal *Enterococcus* populations and the ability of the cheese microbiome to be detected in faecal samples [105,106]. *Enterococcus faecalis* populations in stool samples were significantly increased post-Camembert ingestion, while cheese populations of *L. lactis* and *Ln. mesenteroides* were detected in faeces during the trial [104,105]. Le Roy and colleagues (2022) compared the gut microbial compositions of people who did or did not consume yogurt, using both whole shotgun metagenomic sequencing (*n* = 544) and 16S rRNA sequencing (*n* = 1457, participants overlap with shotgun cohort) [107]. For the larger 16S cohort, yogurt eaters had a significantly higher alpha diversity than those that did not consume yogurt, and seven genera (specifically, four belonging to the *Ruminococcaceae* family, one *Streptococcus*, one unidentified genus belonging to the *Lachnospiraceae* family, and one belonging to the *Christensenellacaea* family) had significantly increased relative abundance [107]. Shotgun sequencing did not mirror these results, instead revealing a significantly positive association between *S. thermophilus* and *B. animalis* and yogurt consumption, whereby greater levels of both species were present in participants who consumed higher quantities of yogurt in a dose-dependent manner, although this increased colonisation appears to be transient [107]. Both of these species are markers of a diet rich in fermented milk products [107].

The ability of fermented foods to impact gut microbial populations is not limited to fermented dairy products. The effect of both fresh and fermented kimchi on the gut microbiome of obese patients was examined in [66]. Although both types of Kimchi caused shifts in microbial populations, namely, an increase in both Proteobacteria and Actinobacteria over the eight week period, certain changes were specific to only one group, such as the increase in Actinobacteria observed in the group receiving fermented kimchi, which was negatively correlated with body fat [66]. Additionally, the fermented kimchi group saw a relative increase in *Bacteroides* and *Prevotella* and a relative decrease in *Blautia* [66]. A study with pasteurized versus unpasteurized sauerkraut intake in IBS patients showed significant improvements in symptoms and a significant change in the composition of the gut microbiota (specifically, reduced operational taxonomic units of the *Clostridiales* order), although the unpasteurized group showed significantly greater numbers of sauerkraut-associated LAB (*Lactiplantibacillus plantarum* and *Levilactobacillus brevis*) than their pasteurized counterparts [108]. Two separate groups investigating the effects of fermented soy milk on healthy adult populations observed significant increases in both bifidobacteria and lactobacilli faecal populations along with decreases in clostridia, with these shifts being partially attributed to the ability of bifidobacteria and not clostridia to utilize certain soybean oligosaccharides [109,110]. Ingestion of Cha-Koji, a green tea fermented with *Aspergillus luchuensis*, and its effects on perturbations in the gut microbiota was investigated in mouse caecal and human faecal samples, showing significant increases in *Clostridium* clusters XI and XIVa (members of which are established butyrate producers), respectively [111]. Faecal *Bifidobacterium* spp. were significantly increased following three weeks of coffee consumption by healthy adult volunteers, although inconsistencies between subjects were observed [112]. Fermented plant extract was found to significantly shift microbial populations in stool samples of mildly hypercholesterolemic patients by increasing bifidobacteria and lactobacilli and decreasing harmful *E. coli* and *Clostridium perfringens* [113]. The effects of raspberry juice fermented with *Lacticaseibacillus casei* on in vitro and in vivo microbiome populations were assessed by Wu and colleagues (2021), with differences being seen in both instances [114]. When the fermented juice was put through artificial digestion followed by colonic fermentation, significant increases in abundance were observed for *Lactobacillus*, *Akkermansia*, *E. coli*, and butyric acid-producing bacteria, while *Bacteroides* and *Ruminococcus* decreased significantly. For their in vivo murine trial, the fermented beverage was freeze dried and fed to male Kun Ming mice in several concentrations, and was found to alter beta diversity and various genera depending on the concentration [114].

Thus, fermented foods have been shown to have the capacity to modify gut microbiome populations, although it can often be unclear as to how these changes are brought about. While this has been shown in the above examples for a myriad of different fermented food types, the results cannot be directly compared due to highly variable parameters, including the use of both healthy and disease models and the various methods by which microbes are quantified. Fermented foods may interfere with the gut microbiome through its own microbiome or through the nutrients present in its matrix. Therefore, in an effort to provide objective evidence that clearly demonstrates whether or not fermented foods can modulate the human gut microbiome, more in-depth and well-defined human feeding studies need to be undertaken. In these studies, the microbiomes of both the fermented food and the human gut need to be established using the most advanced and sensitive tools available in order to permit changes at both the genus and the species level to be determined.

## 5. Nutrients from Fermented Foods That Modulate the Gut Microbiome

In addition to the benefits of fermented foods to the host, certain chemicals present in fermented foods can affect the host’s gut microbiome directly. The production of two types of chemicals in particular have come under scrutiny due to their documented effects on microbial populations, polyphenols and dietary fibre, the latter of which leads to the in vivo production of short-chain fatty acids [115].

Fermentation can lead to increases in polyphenol bioavailability in fermented foods [116,117,118]. Polyphenols are a group of heterogenous chemicals found in plant-based foods comprising both flavonoids and non-flavonoids. In terms of the human diet, flavonoids and phenolic acids comprise the majority of dietary polyphenols and are sought after for their antioxidant properties, and have been demonstrated to directly impact on the gut microbiome [119,120]. A recent study examined the effect of fermentation on polyphenol content in a mix of eight legumes commonly consumed in China [121]. The beans were allowed to ferment either naturally (without added microorganisms) or with added lactobacilli, with both soluble and bound total phenolic content (TPC) measured following 48 h of incubation [121]. While antioxidant capacity differed between the two fermentation methodologies used, significant increases in soluble TPC were seen in both when compared to non-fermented samples, while varying increases were also seen in the bound TPC fraction [121]. Similar studies involving other vegetables have been reported. The free polyphenol content (FPP) of unfermented corn and corn fermented by two strains of fungi (*Agaricus* sp.) was compared; both fermented corns showed an increase in FPP content compared to the unfermented control, with one of the strains having an FPP content 88 times greater than the control [122]. Unsurprisingly, this does not appear to work for all fermented plant products. A study in 1994 investigated polyphenol levels in de-hulled black-gram dhal slurry and found that the fermented product had significantly lower levels of polyphenols in comparison to its raw material, whereby fermentation for 18 h resulted in loss of almost 50% of total polyphenol content [72]. Thus, each fermented plant product should be investigated before claims can be made regarding polyphenol content.

Various studies have shown that polyphenols can impact bacteria found in the gut, although the majority of these studies focus on pathogen inhibition [114,123,124,125]. Tea is a commonly consumed beverage in most countries and advantageously contains a plethora of flavonoids, making it an accessible source of polyphenols [126]. A study in 2006 examined the effects of tea-extracted flavonoids on gut bacteria in vitro and showed that the majority of pathogenic bacteria were inhibited, one of a number of studies that confirms the ability of tea polyphenols to inhibit pathogens [124,126]. On the other hand, the ability of polyphenols to inhibit putatively beneficial bacteria (such as commensal gut bacteria, lactobacilli, and bifidobacteria) has been studied, with many studies concluding that in most cases these microbes are not inhibited [127,128]. While the mechanisms by which polyphenols in general stimulate beneficial gut microbes and inhibit pathogens are unclear, specific properties of each group of microbes have been highlighted as potential explanations. For example, polyphenols can be tolerated by gut microbes and not by pathogens due to their ability to reduce them to less harmful substances; certain gut microbes such as lactobacilli are even capable of utilizing polyphenols as a nutritional substrate [129,130]. In contrast, polyphenols have shown inhibitory effects on virulence factors of pathogenic bacteria, such as the suppression of the *H. pylori* urease enzyme required for its ability to neutralize gastric acid [124]. The microbiota themselves influence the bioavailability of ingested polyphenols through bacterial enzymes (e.g., esterases, demethylases), transforming them into forms capable of absorption through the intestinal wall [123,131].

Wine is a rich source of polyphenols, and manipulation of various factors during grape growth and processing can increase total polyphenol content [132]. Red wine polyphenols have been shown to significantly alter gut microbiota groups, including an increase in overall microbial diversity, and to significantly lower total cholesterol and blood pressure [133,134,135,136]. While Barroso et al. (2017) only observed an overall increase in diversity and could not identify a consensus between healthy wine-consuming individuals, Queipo-Ortuño and colleagues (2012) noted significant increases in *Enterococcus*, *Prevotella*, *Bacteroides*, *Bifidobacterium*, *Eggerthella lenta*, and *Blautia coccoides*–*Eubacterium rectale* groups in similarly healthy individuals who consumed wine once a day, although the dominant groups shifted throughout the duration of the study [133,136]. A potential benefit of wine polyphenol consumption may be the lowering of lipopolysaccharide (LPS) or LPS-producing bacteria, as Clemente-Postigo and colleagues (2013) found a significant increase in both *Bifidobacterium* and *Prevotella* with beverage consumption that was negatively correlated with LPS [134]. Similarly, Moreno-Indias and colleagues (2016) noted that consumption of red wine increased beneficial *Bifidobacterium* and Lactobacillaceae levels, increased the levels of butyrate-producing *Faecalibacterium prausnitzii* and *Roseburia*, and decreased LPS-producing *E. coli* and *Enterobacter cloacae* [135]. Two separate studies examined the effects of quercetin and resveratrol, two polyphenols found in red wine, on gut dysbiosis in rats fed high-fat diets [137,138]. Both studies found that the polyphenols were able to reduce the *Firmicutes/Bacteroidetes* ratio linked with high-fat diets, although Etxeberria et al. found that only quercetin was able to decrease microbial groups previously associated with diet-induced obesity, while Zhao et al. concluded that a combination of quercetin and resveratrol was capable of the same effect [137,138]. As polyphenols can inhibit pathogenic bacteria and potentially benefit advantageous bacteria, the consumption of fermented foods with high levels of polyphenols has the potential to impact gut bacteria.

Short chain fatty acids (SCFA) arise through the catabolism of carbohydrate fibres by microbes. This is important in human nutrition, as microbes indigenous to the colon are capable of this fermentative process and the SCFA thus produced are used as an energy source by colon cells; their formation benefits the human host, as it enables energy to be extracted from an otherwise-indigestible complex carbohydrate [57,139]. Well known examples of SCFA include acetate, propionate, and butyrate; Lactobacillaceae and *Bifidobacterium* are established producers of these valuable components in vitro [115,139]. While SCFA are known to play important roles in host metabolism and the central nervous system, they are important in terms of their impact on the gut microbiota as well [139,140]. As already stated, acetate production by bacteria in any niche will increase the level of acidification of the environment, inhibiting the proliferation of less acid-tolerant bacteria, which is advantageous in the intestines as LAB usually inhibit the growth of pathogens through this mechanism [141]. Acetate has been shown to modulate gut microbiota and reverse gut dysbiosis to a degree [142]. More importantly, SCFA have been demonstrated to stimulate production of mucus (primarily composed of mucin proteins) by host epithelial goblet cells [143]. Mucus coats the intestinal epithelium, with the mucosal layer being thickest in the colon. The majority of gut microbes are found in this layer, and the mucus acts as an energy source for these bacteria [144,145]. A study in rats in 2000 confirmed that the presence of certain SCFA (acetate, butyrate and propionate) increased mucus production in the colon, while a different group in 2003 used a tissue culture model to demonstrate that SCFA stimulated the expression of the mucin-2 protein via prostaglandin production [146,147]. Thus, the presence of SCFA will affect the gut microbiome through influence over pH and mucus concentration.

Certain fermented foods have been shown to contain high levels of readily accessible SCFA [148]. As stated above, vinegar contains high levels of acetate, which is capable of influencing the host gut microbiome, although due to its strong taste direct consumption is difficult for most individuals [149]. SCFA levels in cheeses vary and are often expressed as free fatty acids (FFA), which are important components for the organoleptic properties of cheese and encompass fatty acids of varying lengths [150]. The levels of desirable FFA differ between cheeses, with high levels leading to a rancid off-flavour in varieties such as Cheddar and Gouda [151]. Italian hard cheeses are considered good sources of SCFA, with short and medium-chain fatty acids comprising 25% or more of total triglyceride content in Parmigiano Reggiano and Grana Padano [152]. As stated previously, propionibacteria found in Swiss-type cheeses ferment the lactose into both acetate and propionate SCFA, resulting in products rich in these compounds [153]. A study in 2007 examined the efficiency of *Propionibacterium freudenreichii*, a species commonly used in the production of Swiss-type cheeses, to produce SCFA in rats; one strain in particular was found to significantly enhance the presence of these compounds within the caecum [154].

Thus, fermented foods have the potential to impact gut microbiota by modifying levels of particular compounds within food. Polyphenols have been shown to directly impact the microbial composition of the gut, while SCFA can induce a more favourable environment for the growth of beneficial gut microbes or impact mucus concentration, which acts as both an energy source and binding site for gut microbes.

## 6. Potential of Fermented Food Microbiota to Survive and Modulate the Gut Microbiome

As discussed above, fermented foods contain large and diverse microbiomes, many species of which are to be found within the gut microbiota as well; thus, it is reasonable to propose that fermented foods could be a source of these microorganisms. However, in order for this to occur the fermented food microbiota must have the capacity to survive the environmental stresses of the digestive tract. This requires tolerance to low pH and bile, traits used in the selection of probiotic bacteria. Many groups have investigated the microbiota of fermented foods as a source of new probiotic bacteria, and many have found bacteria that are capable of surviving gastric transit, implying that this property may be common among the microbiota found in fermented foods. Potential to survive gastric transit can be investigated using a range of in vitro models and using either purified bacterial cultures grown in laboratory media or by incorporating the test organism into the model food matrix.

A study in 2011 screened *Lactiplantibacillus plantarum* isolates from both Argentinean and Italian cheeses for potential probiotic characteristics [155]. The most promising isolates were screened for their potential to survive gastric transit by exposure to gastric conditions (pH 2.2 and pepsin) for 90 min followed by 150 min exposure to synthetic duodenal juice (pH 8.0, pancreatin and bile) [155]. Survival rates for simulated gastric juice were based on the percentages of live cell counts after varying time points relative to initial counts prior to exposure, while bile tolerance was expressed as the proportional cell count increase in differing bile concentrations compared to a bile-free control [155]. The *Lactiplantibacillus plantarum* strains tested showed excellent resistance to digestive stresses and on average about 40% tolerance to bile; the data reported were the average of three different bile concentrations, including one more than three times higher than physiological conditions, in which case their survival level in vivo may be much higher [155]. These data imply that these bacteria are capable of surviving gastrointestinal track (GIT) passage, which would then make it possible for them to colonize the gut either transiently or long-term and to interact with the host gut microbiome [155]. In this case, the bacteria were removed from the protective matrix of the fermented food, and should these tests be repeated with intact cheese, the results might differ. It is worth noting that *Lactiplantibacillus plantarum* is present in a myriad of fermented foods made from vegetables, meats and dairy products, and is found in the human intestine as well [156]. While survival remains strain-dependent, *Lactiplantibacillus plantarum* has been investigated by other authors and has been found to survive well under gastric conditions. Haller and colleagues investigated a number of strains both from fermented foods and of intestinal origin and determined that, while the strains from the intestines performed the best under digestive stresses, the *Lactiplantibacillus plantarum* strains survival rates that are used to ferment fruit were broadly similar [157]. A study in 2014 involved isolated strains of different Lactobacillaceae genera responsible for fermentation of Croatian white cabbage for the production of sauerkraut [158]. While initial LAB counts were approximately 5.95 log CFU/mL, the group identified four *Lactiplantibacillus plantarum*, one *Ln. mesenteroides* ssp. *mesenteroides*/*dextranicum*, and one *Levilactobacillus brevis* strain, all of which were capable of surviving simulated digestion, including low pH, bile salts and digestive enzymes, at rates of above 5 log CFU/mL, resulting in only approximately 1 log CFU/mL reduction following exposure to simulated GIT conditions [158]. Mishra and Prasad exposed *Lacticaseibacillus casei* strains from fermented dairy products to low pH (1.0, 2.0 and 3.0) and high bile (1%–2%), revealing three acid-tolerant strains (although no strains survived at pH 1.0) and strains with varying levels of bile tolerance [159]. While the *Lacticaseibacillus casei* strains appear to be less tolerant of simulated digestion, it should be taken into account that they would normally be found in a fermented food matrix, and the bile concentrations used were very high in comparison to physiological conditions [159]. While most reports to date have focused on studying bacterial strains isolated from fermented foods, a recent study exposed Cheddar cheese to simulated gastric digestion prior to isolation of NSLAB in an effort to understand whether significant populations of this group of bacteria naturally present in most cheeses could potentially survive gastric transit [160]. The observation from this work was that a population of 10^7^ CFU/g could survive simulated gastric transit, a sufficient number to potentially impact the gut microbiome. This research demonstrates the in vitro potential of Lactobacillaceae strains from fermented foods to survive gastric transit, potentially allowing them to influence the gut microbiota. However, the importance of the fermented food itself to protect the bacteria during digestion needs more consideration.

In other cases, fermented foods are used as a delivery vehicle for probiotic strains that may not be autochthonous for that environment. In 2005, a group investigated the effectiveness of fermented olive varieties (in particular, their skin) at carrying the established probiotic *Lacticaseibacillus paracasei* IMPC2.1 into the intestines of five human subjects, who were given 10–15 olives coated with 10^9^–10^10^ CFU of lactobacilli [161]. Faecal samples were taken prior to consumption of olives (subjects were told not to ingest any lactobacilli-containing edibles for one week prior to the trial) and at days 10 and 15 post-consumption [161]. While the *Lacticaseibacillus* strain was absent in the faecal samples prior to ingestion of the olives, it was detected in four of five of the subjects post-consumption using a combination of Vancomycin-containing agar plates and PCR-Amplified Ribosomal DNA Restriction Analysis (ARDRA) [161]. Therefore, this study indicates that the olives were effective fermented food vehicles for safely delivering probiotics through gastric transit [161]. A study by Saxelin and colleagues in 2010 investigated the potential survival of the probiotic strains *Lacticaseibacillus rhamnosus* GG and LC705, *Propionibacterium freudenreichii* subsp. *shermanii* JS, and *Bifidobacterium animalis* subsp. *lactis* Bb12 when administered as either capsules or interred in yogurt or cheese [162]. In this study, while no matrix effect was seen for either *Lacticaseibacillus rhamnosus* strain based on counts in faeces, both the *P. freudenreichii* and *B. animalis* strains showed higher survival levels when administered in the yogurt matrix, although counts in the other matrices were still high (log10) [162]. This study indicates that certain matrices may be more suited to certain probiotics than others.

In 2008, yogurt and low-fat cheese matrices were compared as barriers against acid exposure for the probiotic *Lacticaseibacillus casei* 334e, with the added stage of refrigerating both matrices for time periods reflecting their shelf-life (up to three weeks and three months for the yogurt and cheese, respectively) in order to test their ability to sustain probiotics during storage [163]. The *Lacticaseibacillus* strain was inoculated into both fermented foods during the manufacturing process at 10^7^ CFU/g and remained at this viable count after storage in both cases [163]. This study demonstrated that the low-fat cheese was the better option, as numbers of *Lacticaseibacillus casei* remained higher following exposure to pH 2.0, stabilizing at approximately 10^4^ CFU/g after 120 min of exposure in cheese, as opposed to less than 10^1^ CFU/g in yogurt after only 30 min [163]. A recent study used the INFOGEST 2.0 simulated gastric digestion model to investigate the survival of *Lacticaseibacillus paracasei* and *Lacticaseibacillus rhamnosus* strains embedded in two dairy matrices, Cheddar cheese and fermented milk [164]. The bacterial strains were added to the two fermented dairy foods during manufacture and thus were distributed, embedded, and protected as would occur naturally. Cheese offered greater protection to the two strains than fermented milk, with log reductions in the range 0.88–1.93 as opposed to 2.36–3.00 following treatment by INFOGEST 2.0. It is relevant to note, however, that 5.38–6.85 log CFU/g of the test bacteria survived the treatment from both dairy matrices, indicating that both are potentially suitable carrier foods for delivery of lacticaseibacillii strains to the intestine. *Limosilactobacillus reuteri* LR92, a strain with known potential probiotic properties, was tested for its ability to survive gastric transit using a fermented vegetable/fruit blended beverage as a matrix [165]. Unlike the study by Sharp et al. in 2008 [163], storage in this manner resulted in varying levels of probiotic survival when the blend was run through acid exposure and bile salt tolerance assays in parallel [165]. While initial levels were about 10 log CFU/mL in both, these decreased by about 5 log cycles when exposed to pH 1.5 for 2 h, and only decreased by about 1 log cycle when exposed to bile salts at pH 7.4 for 150 min [92]. Thus, the fermented juice blend appears to be better at protecting against stresses present in the small intestines than against those present in the stomach [165]. Fermented vegetable/fruit beverages have the added advantage of appealing to vegetarian consumers and those with lactose intolerance issues [166]. Therefore, fermented foods can be successfully deployed as vehicles to sustain high numbers of probiotic bacteria during digestion in order to have an effect on the gut microbiota and health.

Few studies that examine the microbial consequences of ingesting fermented foods have been reported in the literature, and those that are available are usually grouped together with a diet (such as cheese) or with the focus on a probiotic of interest that is present already or more likely has been added in high cell numbers, with the fermented food acting as a vehicle for said strain. It has previously been established that diets dominated by specific macronutrients and deficient in others effect the composition of the gut microbiome, with enterotypes (defining microbiotas based on their dominating taxa) developing when these eating patterns are long-term [167,168]. In general, a diet high in animal protein leads to high numbers of *Bacteroides* and other groups with high bile tolerance being present in the intestines of the human host due to the greater concentration of bile excreted to deal with animal fat digestion, while a high-carbohydrate diet is associated with the *Prevotella* enterotype [167,169]. Protein fermentation by gut microorganisms has been linked with the production of precursors for toxic and in some cases carcinogenic substances as well as those associated with immune syndromes such as inflammatory bowel disease [168]. A study performed in Japan by Odamaki et al. attempted to observe the effect of a yogurt drink that had been fortified with the probiotic *Bifidobacterium longum* on a cohort of volunteers who were put onto an animal-based diet for five days, followed by a ‘recovery’ balanced diet for fourteen days [170]. Three different groups were established, those who were given the fortified yogurt for the duration of the trial (YAB group), those who were only given the yogurt during the fourteen-day ‘recovery’ period (YB), and those who did not receive any yogurt (CTR) [170]. Following the animal-based diet, the gut microbiome of both YB and CTR groups experienced an increase in the genera *Bilophila*, *Odoribacter*, *Dorea*, and *Ruminococcus*, all of which are associated with either a metabolic disorder (Metabolic Syndrome, Inflammatory Bowel Disease, or Crohn’s Disease) or other illness, including colon cancer [171,172,173,174]. Additionally, the relative abundance of *Bifidobacterium* decreased in these two groups [170]. However, the gut microbiomes of the YAB group did not experience the same alterations in abundance as the previously-mentioned bacterial groups (apart from the *Dorea* group), indicating that the probiotic yogurt may have had a stabilizing capacity on the gut microbiota, although the authors stated that this may be attributed to the higher carbohydrate levels ingested in the YAB group [170]. As the strain of *Bifidobacteria* used in this study was known to produce bile salt hydrolase enzymes, it was proposed that this might limit the growth of the bile-metabolizing groups *Bilophila* and *Odoribacter* [170,175,176]. While this study did indicate a degree of crosstalk between a strain from the fermented food and the gut microbiota, the other starter and possible adventitious strains present in the yogurt were not mentioned as possible contributing factors.

In 2011, McNulty and colleagues used controlled conditions to ascertain the effects of a specific set of five known bacteria belonging to the family Lactobacillaceae and the genera *Bifidobacteria*, *Lactococcus* and *Streptococcus* on the microbiome of seven sets of female monozygotic human twins while, in a parallel study, gnotobiotic mice were seeded with a model human gut microbiome of known species the genomic sequences of which were available [177]. The purpose of using such a model was to lower the potential variability associated with genetic makeup, environment, and diet. With regard to the human trials, monozygotic twins were used due to their near-identical genetic makeup and the fact that their early dietary and environmental experiences are very similar to each other. Additionally, unique microbial subsets have been identified in monozygotic twins that result from long-term exposure to similar environmental factors, reinforcing their usefulness when investigating the effect of food microbes on existing gut populations [178]. The five bacteria of interest were used as starter cultures to produce a fermented milk beverage which was given to the individuals, who then supplied faecal samples for analysis [177]. The mice were not fed the fermented milk beverage, and instead orally received the five strains used to culture it, negating any protective effects the fermented food may have afforded the microbes [177]. In any case, the beverage did not alter the human gut microbiome to a significant extent, although the *Bifidobacterium* strain was recovered in the highest amounts from the faecal samples in comparison to its companion starter strains [177]. The authors reported that the five strains were no longer detected following cessation of fermented beverage consumption, indicating that detection of the strains of interest during this period was a result of the beverage and not related autochthonous gut strains [177]. However, the most interesting results were the metatranscriptome profile changes that occurred in both the mice and human subjects, the majority of which resulted in upregulation of genes involved in plant polysaccharide metabolism [177]. This may indicate that the microbiome of the fermented milk impacted the gut microbiome despite not being retained once consumption of the beverage stopped. Thus, the food and gut microbiome exhibited a degree of interaction that allowed a change in gene expression.

Another study reported the impact of yogurt containing deep sea water (DSW, a food supplement) on the gut microbiome of clinical subjects [179]. This trial contained three groups: one which received the DSW-fortified yogurt, one which received non-fortified yogurt, and a control group that received only normal water [179]. In terms of microbiome perturbation, investigation of the intestines of the mice revealed an increase in LAB and total bacteria counts in both yogurt groups that was significantly higher than the water-only control group [179]. While this does seem to indicate that the fermented food had an impact on the gut microbiome, it is worth mentioning that the actual strains used as starter cultures in the yogurt (a *Lactiplantibacillus pentosus* strain and a *Pediococcus pentosaceus* strain) were not tracked through the experiment, and it is unclear whether these specific strains comprised a proportion of the LAB counts or whether they were even able to survive the initial gastric transit [179]. Additionally, the addition of an unfermented milk control would have been useful to confirm that the microbiome changes were not contributed by nutrients already present in the milk.

Although the relative abundance of fermented food microbiota are not uniform, the gut microbiome is exposed to and interacts with these allochthonous bacteria following ingestion either transiently or on a more long-term basis [180,181]. The above studies suggest that it is possible for the microbiome of fermented food products to have varied effects on the host gut microbiome, with many of the resident bacteria present in the food microbiome associated with beneficial responses in the gut and able to harbour potentially probiotic microorganisms. However, in order to establish with greater confidence that the microbiome of fermented foods can survive gastric transit, more consistent and standardized models of the human gastrointestinal tract, such as the INFOGEST 2.0 model, need to be applied and the results confirmed in human feeding studies.

## 7. Conclusions

Fermented foods have an important place in human history, and while their primary function was originally shelf life extension of seasonal foods, health benefits associated with their consumption have long been recognized [6]. Almost all the primary foodstuffs consumed by humans can be subjected to fermentation [182]. Many of these have regional origins linked to the prominent foods produced in different localities. Local modification and adaptation of the fermentation process has greatly added to the diversity of fermented foods, as in the case of cheese; while cheese is made using milk from a limited number of mammal species, variations in fermentation technology have resulted in >1000 cheese varieties [15]. As described above and summarized in Table 1, there are six primary food fermentation pathways, and the starter microorganisms responsible for these are well characterized. However, a diverse range of non-starter microbes are associated with most fermented foods, the full extent of which is only now becoming apparent with the application of high-throughput DNA sequencing technologies [4]. Recently, a freely accessible online database has been established that functions as an archive for the annotated genomic, metagenomic, transcriptomic, and metataxonomic information of microbes associated with fermented foods, enabling evaluation of individual strains as potential starter cultures [183]. This tool has the potential to reveal the full diversity of fermented food-associated microbes, and has useful practical applications in the fermented food industry.

As originally documented by Metchnikoff, fermented foods are being investigated for their ability to exert health benefits. There is growing scientific evidence to support this contention, including data demonstrating that fermented foods can be more easily digested due to partial protein digestion during fermentation as well as indications that they can be enriched in certain vitamins and antioxidants. It has been demonstrated that fermentation can result in the release of bioactive peptides, for example the well documented ACE inhibitory peptides and the production of bacterial EPS, which can help to reduce cholesterol. Controlled human clinical trials are the gold standard used to confirm benefits associated with foods or pharmaceuticals; however, while a number of studies have demonstrated health benefits associated with fermented foods (for a detailed review, see Marco et al., 2017 [57]), more such studies are required.

The human gut microbiome has received much attention in recent years, with increasing evidence that it impacts both physical and mental health [184,185] and that many metabolic disorders are associated with disruption to the gut microbiome [186,187]. Lifestyle, including diet, can impact the gut microbiome, and there is increasing interest in the potential of exploiting foods to positively modulate it [187,188,189,190]. Fermented foods offer an opportunity to positively impact the gut microbiome by either (I) providing nutrients to promote or inhibit members of the gut microbiome, or (II) members of the food microbiome establishing residence in the gut and/or interacting with the resident gut microbiome. However, few studies to date have specifically investigated the impact of fermented food consumption on the gut microbiome. Certain bioactive compounds produced by the microbes within food, including polyphenols and SCFAs, can have beneficial effects when consumed. Certain microbial strains found in food are capable of surviving digestion, and fermented foods can act as useful vehicles to carry probiotic strains safely into the gut. However, there are very few studies which focus on the potential of microbiota from fermented foods to impact the gut microbiome; one study investigating the consequences of yoghurt consumption observed higher levels of LAB associated with the gut contents [179]. In addition, most studies of the gut microbiome focus on the microbiota of the large intestine, due primarily to ease of sampling; however, it is likely that fermented foods have more influence on the microbiome in the small intestine, as this location exhibits a higher proportion of LAB and other more aerotolerant bacteria often associated with fermented foods. The possibility that fermented foods impact the gut microbiome is intriguing, and merits more study and additional efforts to include investigations of the small intestine.

## Figures and Tables

**Table 1 nutrients-14-01527-t001:** A summary of the primary fermentation pathways of relevance to fermented foods, including the main fermentation end products and the microorganisms responsible for their production.

Fermentation Type	Substrate	End Products	Microorganisms Responsible	Reference
Lactic Acid *	Sugar			
Homo lactic		Lactic acid	*Lactococcus lactis*	[18]
			*Streptococcus thermophilus*	
			*Lactobacillus delbrueckii* subsp. *bulgaricus*	
			*Lactobacillus acidophilus*	
			*Lactobacillus helveticus*	
			*Pediococcus*	
			*Enterococcus*	
Hetero lactic		Lactic acid, ethanol, CO_2_	*Leuconostoc*	[18]
			*Fructilactobacillus sanfranciscensis*	
			*Levilactobacillus brevis*	
			*Limosilactobacillus fermentum*	
			*Limosilactobacillus reuteri*	
			*Lacticaseibacillus casei*	
			*Lactiplantibacillus plantarum*	
			*Latilactobacillus curvatus*	
Ethanol	Sugar	Ethanol, CO_2_	*Saccharomyces cerevisiae*	[19]
			*Zymomonas mobilis*	
Acetic Acid	Ethanol	Acetic acid	*Acetobacter*	[20]
			*Komagataeibacter*	
Propionic Acid	Lactic Acid	Propionic acid, acetic acid, CO_2_	*Propionibacterium freudenreichii*	[21]
			*Propionibacterium jensenii*	
			*Propionibacterium thoenii*	
			*Propionibacterium acidipropionici*	
			*Propionibacterium cyclohexanicum*	
Citric Acid	Citric Acid	Acetate, Formate, Ethanol, 2,3-butanediol, Diacetyl, Acetoin, CO_2_, Lactate	*Lactococcus lactis* subsp. *lactis* biovar *diacetylactis*	[22]
			*Leuconostoc* **	
			*Enterococcus* **	
			*Lactobacillus* **	
			*Oenococcus oeni*	
Malolactic	Malic Acid	Lactic acid, CO_2_	*Oenococcus oeni*	[23]
			Lactobacillaceae ***	
			*Pediococcus* **	

* Certain LAB, referred to as facultative heterofermentative LAB, can ferment by either the homo- or heterolactic fermentation pathway depending on environmental conditions or substrate availability. ** Certain members of this genus. *** Certain members of this family.

## Data Availability

Not applicable.

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
