# Peer review of "Fermented Foods, Health and the Gut Microbiome"

_nutrients, 2022, doi:10.3390/nu14071527_

Round 1

Reviewer 1 Report

The present review discussed the effects of fermented foods on gut health and gut microbiota composition. Authors put hard efforts to clarify the link between fermented foods and human health, however, the review is too long to make us catch the key information. My detailed information is listed below.

The overall structure of this review looks fine, but I highly recommend adding subheadings for each section.

The review is too long, I think authors needs to control their words within 10,000 words.

The figures/tables are too few. Figure1, there are too many reviews describing the human gut microbiota compositions. You need to draft the figure the figure directly linking the fermented foods on the gut microbiota.

I think you need to explain the differences/compositions changes after fermentation. What are their benefits to the body?

Relating the Potential of the Fermented Food Microbiota to Survive and Modulate the Gut Microbiome, you need to add a table to carefully summarize them.

Minor,

Abstract, please discuss the significance of this review.

Table, M/os Responsible means what?

Author Response

Reviewer 1:

The present review discussed the effects of fermented foods on gut health and gut microbiota composition. Authors put hard efforts to clarify the link between fermented foods and human health, however, the review is too long to make us catch the key information.

My detailed information is listed below.

The overall structure of this review looks fine, but I highly recommend adding subheadings for each section.

We agree with this suggestion so in response to the comment from the Editor we have elevated the subsection in the original Section 6 to full Sections 4, 5 and 6. We have also added four subheadings to Section 3 in accordance with the recommendation of this Reviewer.

The review is too long, I think authors needs to control their words within 10,000 words.

We have removed Sections 4 and 5 of the original manuscript in an effort to address the length of the review while also focusing on the prime objective of the review which is to explore the health benefits of fermented foods and their potential to impact on the human gut microbiome.

Figure1, there are too many reviews describing the human gut microbiota compositions.

As indicated above we have removed Section 2 so Figure 1 has also been removed. 

You need to draft the figure the figure directly linking the fermented foods on the gut microbiota.

We gave this suggestion a great deal of consideration as we agree with the Reviewer that such a figure would provide a very useful summary of the various studies reported in the literature. However, as we outline in the text of the manuscript (lines 538-544), while the overall observations are that fermented foods “have the capacity to modify gut microbiome populations” the “results cannot be directly compared due to highly variable parameters, including the use of both healthy and disease models, and the various methods by which microbes are quantified”. In addition, we also stated that the changes observed are generally “overall shifts in microbial populations, and do not necessarily reflect the microbial content of the fermented foods in question” (lines 428-430). Therefore, after careful consideration we concluded that preparing such a figure could provide an over simplified image of the complex and dynamic observations reported in the literature and that it was better to summarise and discuss this in the text.

However, we do think the point raised by the Reviewer is very valid, and therefore we also added the statement “Therefore, in an effort to provide objective evidence that clearly demonstrates whether or not fermented foods can modulate the human gut microbiome, more in depth and well defined human feeding studies need to be undertaken. In these studies the microbiome of both the fermented food and the human gut need to be established using the most advanced and sensitive tools available so that changes to genus but also species level can be determined” at lines 544-549 to highlight that more information is required to establish a clear and consistent conclusion as to the potential of fermented foods to modulate the human gut microbiome.  

I think you need to explain the differences/compositions changes after fermentation. What are their benefits to the body?

We agree with this suggestion and in consequence have a new subsection “3.2 Transformation in Foods Arising from Fermentation” that highlight some of the key changes that have potential health benefits that come about in food as a consequence of fermentation.  

Relating the Potential of the Fermented Food Microbiota to Survive and Modulate the Gut Microbiome, you need to add a table to carefully summarize them.

In the text of the manuscript we describe the research undertaken on this topic. The difficulty with formulating these experiments into a table format are that the experimental approaches and methodologies vary significantly across the different reports. Some use purified bacteria exposed to various simulated gastric conditions, others expose specific strains embedded within different food matrices to simulated gastric conditions, while others use specific strains embedded with different food matrices in human or animal feeding studies. While we see merit in the suggestion from the Reviewer, summarizing such varied studies in a table format is difficult.

However, this suggestion by the Reviewer was very insightful and in consequence we added the text “However, to establish with greater confidence that the microbiome of fermented foods can survive gastric transit more consistent standard models of the human gastrointestinal tract, such as the INFOGEST 2.0 model need to be applied and the results confirmed in human feeding studies” at lines 868-871.   

Minor,

Abstract, please discuss the significance of this review.

We have edited the abstract as suggested. We have removed reference to (i) “culture” and replaced it with “diet” in the first line of the abstract, (ii) “processing conditions”, which are not really covered in this manuscript and (iii) replaced the last sentence in the original abstract with the statement “This review clearly shows that fermented foods can affect the gut microbiome, in both the short and long term, and should be considered an important element of the human diet”.  

Table, M/os Responsible means what?

Corrected

Reviewer 2 Report

The publication deals with an important aspect of the link between fermented foods and human health and gut microbiota. A lot of articles have been published on this topic recently, so the reviewer expected to receive the most recent information, not an outdated. Meanwhile, out of 205 cited references, only 28 are from the last 5 years, including only 9 from 2020-2021, no citation from 2022.  The method of citing references does not comply with the Nutrints journal guidelines - numbers instead of authors' names should be included in the text, what may suggest that the manuscript was originally prepared for another journal and citation was not changed.

Table 1 is unclear. It is not known where the data in table 1 comes from - no citation. Why enter the abbreviation M/os? No substrates for homo and hereto lactic; misplaced asterisks for family and genus - e.g. Lactobacillus and Enterococcus are not families but genera, etc.

It would be much better to present several biochemical pathways  listing the substrates, intermediates and end products, and enzymes, instead of Table 1 and long text in p. 2.2.

3 species listed in line 207-208: L. plantarum, L. casei and L. curvatus are not included in table 1.

Lack of references concerning indicated particular species and genera (201-208, 285-296).  

Little attention was paid to the dietary differences in populations living in the different geographic regions:  Europe, Asia, Africa and Americas, and to changes in nutrition under the influence of urbanization and globalization, were indicated.

Whether it is necessary to mention side by side – Bacteroides and Bacteroides uniformis – lines 784-785.

What means “negative microbial groups” – line 800.

Instead of sing “    ÍŒ  “  should be written “about” lines: 946-947.

Author Response

The publication deals with an important aspect of the link between fermented foods and human health and gut microbiota. A lot of articles have been published on this topic recently, so the reviewer expected to receive the most recent information, not an outdated. Meanwhile, out of 205 cited references, only 28 are from the last 5 years, including only 9 from 2020-2021, no citation from 2022.

In the revised document the reference list has been updated. Arising from omitting the original sections 4 and 5 and adding new material to some of the other sections the revised document contains 190 citations, 41 of which relate to literature published during the period 2017-2019, 18 relating to the period 2020-2021 with 4 from 2022 plus 1 in press.  

The method of citing references does not comply with the Nutrints journal guidelines - numbers instead of authors' names should be included in the text, what may suggest that the manuscript was originally prepared for another journal and citation was not changed.

Method of citation and reference listing were changed and now presented as per the Author Guidelines  

Table 1 is unclear. It is not known where the data in table 1 comes from - no citation.

This table is original prepared by the authors for this manuscript. We have added relevant citations in the revised Table 1 as suggested.   

Why enter the abbreviation M/os

Corrected

No substrates for homo and hereto lactic.

We stated in the text that “The substrate for both lactic acid and ethanol fermentation are sugars available in the food material” and to strengthen this in response to the comments by the Reviewer we have added the text “for example but not limited to lactose in milk or glucose from the breakdown of starch in plant products” (line 129-130).

Misplaced asterisks for family and genus - e.g. Lactobacillus and Enterococcus are not families but genera, etc.

Corrected

It would be much better to present several biochemical pathways listing the substrates, intermediates and end products, and enzymes, instead of Table 1 and long text in p. 2.2.

While we understand this suggestion, when we were preparing this manuscript we decided not to include biochemical pathways as we did not want to go into too much detail on the biochemistry of food fermentation in this review where the emphasis was on the potential health benefits of such foods. In addition, there are subtle differences in some of the pathways (depending on the organisms involved and environmental conditions) which we felt were better addressed by mention in the text of the document rather than by including images of the pathways.

3 species listed in line 207-208: L. plantarum, L. casei and L. curvatus are not included in table 1.

These three species are now included in Table 1 as suggested.

Lack of references concerning indicated particular species and genera (201-208, 285-296).

To address this reference number 26 (Tamang 2014) is added at lines 166 and 170 to address comment regarding lines 210-208 in the original document; and reference number 41 (Tamang et al 2017) is added at line 247 to address the comment regarding lines 285-296 in the original document.   

Little attention was paid to the dietary differences in populations living in the different geographic regions:  Europe, Asia, Africa and Americas, and to changes in nutrition under the influence of urbanization and globalization, were indicated.

We agree with this comment; however, it is a very broad topic and dietary differences and their potential to impact on health are outside the scope of this review. However, we think that the point made by the Reviewer is very valid so we have added some text (Line 120-121) and a new reference number 2 (Tamang et al 2020) which is an excellent review that deals with the global diversity in fermented foods and their different methods of manufacture from traditional spontaneous fermentation to industrial production using defined strain starter cultures.  

Whether it is necessary to mention side by side – Bacteroides and Bacteroides uniformis – lines 784-785.

Corrected

What means “negative microbial groups” – line 800.

We have addressed this by deleting reference to “negative microbial groups” and replacing it with “microbial groups previously associated with diet induced obesity”

Instead of sing “    ÍŒ  “  should be written “about” lines: 946-947.

Have corrected this and also similar corrections at other locations where “   ÍŒ” was used in the original manuscript  

Round 2

Reviewer 1 Report

I recommend the acceptance of this paper. No comments further.